# Efficient Advanced Oxidation Process (AOP) for Photocatalytic Contaminant Degradation Using Exfoliated Metal-Free Graphitic Carbon Nitride and Visible Light-Emitting Diodes

**Adeem Ghaffar Rana [1,2]**, **Minoo Tasbihi [3]**, **Michael Schwarze [3]** and **Mirjana Minceva [1,\*]**

1    Biothermodynamics, TUM School of Life Sciences Weihenstephan, Technical University of Munich, Maximus-von-Imhof-Forum 2, 85354 Freising, Germany; adeem.rana@tum.de
2    Department of Chemical, Polymer and Composite Materials Engineering, University of Engineering and Technology (UET), Lahore 39161, Pakistan
3    Department of Chemistry, Technische Universität Berlin, Str. des 17. Juni 124, 10623 Berlin, Germany; minoo.tasbihi@tu-berlin.de (M.T.); michael.schwarze@tu-berlin.de (M.S.)
*    Correspondence: mirjana.minceva@tum.de; Tel.: +49–8161–71–6170

**Abstract:** The photocatalytic performance of metal-free graphitic carbon nitride ($g$-$C_3N_4$) was examined using visible light-emitting diodes (LEDs). A comparative and parametric study was conducted using the photocatalytic degradation of phenol as a model reaction. The $g$-$C_3N_4$ photocatalyst was synthesized from melamine using thermal condensation, followed by a thermal exfoliation that increases the catalyst surface area from 11 to 170 $m^2$/g. Different characterization techniques, namely X-ray powder diffraction, X-ray photoelectron spectroscopy, nitrogen adsorption using the Brunauer–Emmett–Teller method, ultraviolet-visible (UV–vis) spectroscopy, transmission electron microscopy, photoluminescence spectroscopy (PL), and zeta potential analysis, were used to characterize the photocatalyst. A comparison of the photodegradation experiments conducted with a full-spectrum xenon lamp and a custom-made single-wavelength LED immersion lamp showed that the photocatalyst performance was better with the LED immersion lamp. Furthermore, a comparison of the performance of exfoliated and bulk $g$-$C_3N_4$ revealed that exfoliated $g$-$C_3N_4$ completely degraded the pollutant in 90 min, whereas only 25% was degraded with bulk $g$-$C_3N_4$ in 180 min because the exfoliated $g$-$C_3N_4$ enhances the availability of active sites, which promotes the degradation of phenol. Experiments conducted at different pH have shown that acidic pH favors the degradation process. The exfoliated $g$-$C_3N_4$ has shown high photocatalytic performance in the photodegradation of other phenolic compounds, such as catechol, m-cresol, and xylenol, as well.

**Keywords:** $g$-$C_3N_4$; photocatalysis; LEDs; wastewater treatment; phenol

## 1. Introduction

Phenol and its chemical derivatives are widely used in pharmaceutical, textile, pesticide, and petrochemical industries; therefore, they are characteristic pollutants in the effluent from these industries. Polluted water is harmful to human health and the ecosystem [1,2]. In this context, several efforts have been made to completely remove phenolic contaminants from wastewater using conventional methods, including adsorption [3,4], distillation [5], biological treatments [6], and chemical oxidation [7]; however, these techniques have their demerits. Heterogeneous photocatalysis has proven to be an effective and feasible method for removing organic contaminants, producing $CO_2$ and mineral salts as byproducts [8]. Many research efforts have been devoted to the development of advanced oxidation processes (AOPs) using ultraviolet (UV) and visible light [9]. Sunlight has limitless energy, which can be used for the activation of photocatalysts; however, it has a wide wavelength range. In this regard, visible light-emitting diodes (LEDs) offer the selectivity and flexibility of a specific wavelength, the possibility of designing the reactor, and cost-efficiency for achieving controlled irradiation [10,11].

Many semiconductor materials reported in the literature, for example, titanium dioxide ($TiO_2$) [12] and zinc oxide (ZnO) [13], are promising because of their availability, nontoxicity, and photostability. However, there are numerous drawbacks associated with these materials, such as (i) a high bandgap energy (3.3–3.5 eV), which limits their utilization to only the UV region of sunlight, i.e., 4% of the solar energy, and (ii) a high rate of recombination of electrons and holes, which is detrimental to the efficiency of the material [14–17].

Graphitic carbon nitride ($g-C_3N_4$) is a recently reported polymeric semiconductor consisting of heptazine networks, which has attracted much attention because of its wide range of photocatalytic applications [11]. It exhibits chemical and thermal stability along with promising optical and electronic semiconductor properties. The additional advantages of $g-C_3N_4$ are its easy synthesis and suitable band gap (2.7 eV) [18,19] for the efficient use of solar light approximately 430 nm with sufficient energy to conduct oxidation and reduction applications, for example, $H_2$ production, $CO_2$ reduction, and degradation of organic pollutants [20–24].

Generally, $g-C_3N_4$ can be synthesized via thermal decomposition of nitrogen-rich precursors, such as melamine, urea, thiourea, cyanamide, or dicyanamide, which results in the formation of two-dimensional tri-*s*-triazine sheets [25]. The main bottleneck limiting the practical application of pristine $g-C_3N_4$ is the fast electron-hole recombination in the excited state, which results in a limited activity because of a low surface area [15]. In the design of photocatalysts, different strategies have been used to improve the surface area, such as templating, doping, or exfoliation [26]. Among these strategies, thermal exfoliation has attracted particular attention as a fast, easy, and efficient method to produce thin-layered $g-C_3N_4$ of high quality. During the heating process of this procedure, oxygen-containing functional groups of $g-C_3N_4$ decompose and generate large amounts of gas, creating sufficient pressure to overcome van der Waals interactions and further expand the layers to form porous frameworks [10]. More importantly, thermal exfoliation has been demonstrated as a convenient way to enhance the surface area, to introduce pores, and to tune the texture of the material [26].

In photocatalytic contaminant degradation, very reactive species are formed during irradiation that decomposes the contaminant, in the ideal case to $CO_2$ and water. The basic principle of conventional AOP involves the production of hydroxyl radicals (HO•) as reactive molecules, which can be generated from hydrogen peroxide ($H_2O_2$). Meanwhile, as a difference to conventional AOPs, the proposed mechanism of the photocatalytic degradation of organic pollutants using metal-free $g-C_3N_4$ suggests the in situ production of reactive oxygen species via oxygen reduction so that no further auxiliaries are required [10,27].

The main objective of the present study was to carry out a comparative study between an LED and a xenon immersion lamp using exfoliated $g-C_3N_4$. Although LEDs have found their way into photocatalysis for water splitting [28] or contaminant degradation [10], comparative reports between light sources are rare. This is, to the best of our knowledge, the first comparative report of its kind for exfoliated $g-C_3N_4$ using two different lamps. Using exfoliated $g-C_3N_4$, an LED immersion lamp, and phenol as a model pollutant, the effect of process parameters, such as pH, catalyst concentration, pollutant concentration, air flow, and exfoliation on the performance of the catalyst, was evaluated. The study was extended to the photodegradation of other phenolic compounds, including m-cresol, catechol, and xylenol.

## 2. Results and Discussion

### 2.1. Photocatalyst Characterization

The morphology of bulk and exfoliated $g-C_3N_4$ was analyzed by TEM. Figure 1 presents the selected micrographs. The structure of bulk $g-C_3N_4$ shows continuous and closed stacks with irregular aggregates. After the exfoliation, the material transformed into a thin nanosheet. The layer thickness was reduced, and the large layer split into a small nanosheet

because of exfoliation [15,29]. It can be inferred from these results that the exfoliation enabled the formation of porous nanosheets with increased specific surface area in g-C$_3$N$_4$ [30].

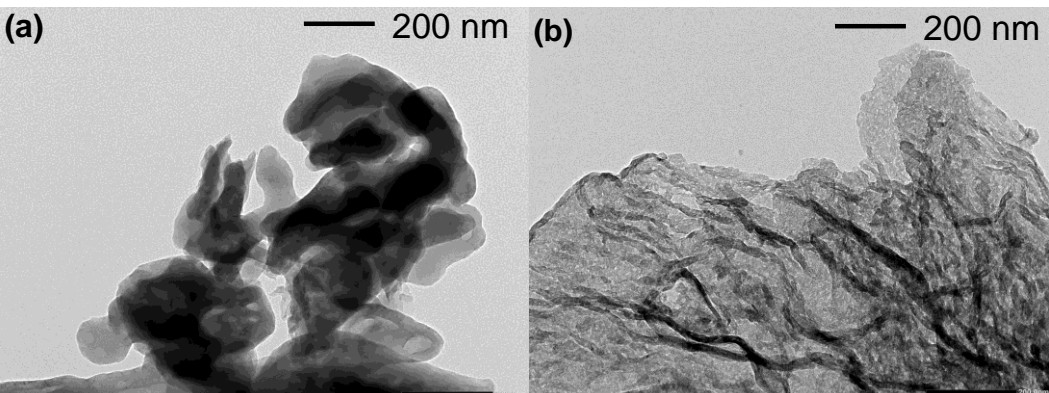

**Figure 1.** Transmission electron microscopy images of bulk (**a**) and exfoliated (**b**) graphitic carbon nitride (g-C$_3$N$_4$).

Specific surface areas were measured to reveal any change in the structural features of carbon nitride before and after the exfoliation. The surface area of exfoliated g-C$_3$N$_4$ was significantly higher than that of bulk g-C$_3$N$_4$ (170 and 11 m$^2$/g, respectively), and the pore size distribution was also increased for the former, according to the TEM results. Hydrogen bonding and van der Waals forces were the main interactions contributing to the stacking between g-C$_3$N$_4$ layers, and their removal caused the exfoliation into thinner layers. This process led to an increase in the surface area of the material.

Figure 2 shows the N$_2$ adsorption-desorption isotherms and BJH pore size distributions of bulk and exfoliated g-C$_3$N$_4$. Both samples show a hysteresis loop of type IV, suggesting the existence of a large number of pores. These characteristic curves for mesoporous materials stem from the difference in the curvatures of the meniscus during adsorption and desorption with a capillary condensation. The pore size distribution curves of both materials show a maximum at a radius of 1.9 nm, which means that the materials were mesoporous [31,32].

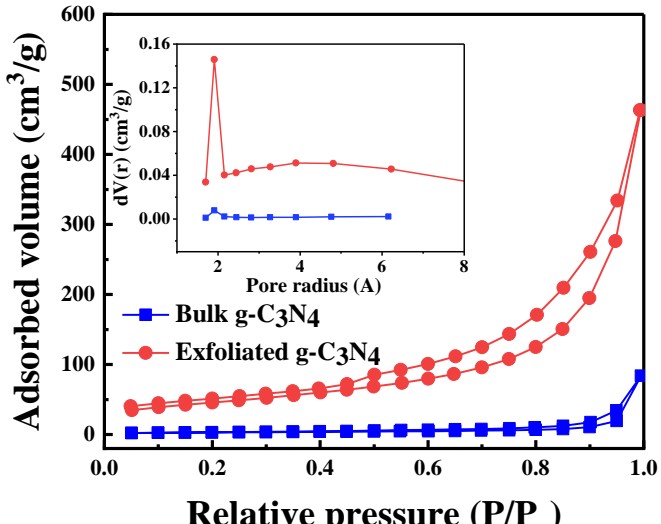

**Figure 2.** N$_2$ adsorption-desorption isotherms of bulk and exfoliated g-C$_3$N$_4$. The inset shows the corresponding Barrett–Joyner–Halenda pore size distribution curves of the samples.

To study the optical properties, bulk and exfoliated g-C$_3$N$_4$ were characterized using UV–vis and PL spectroscopy. Figure 3a presents the PL spectra of bulk and exfoliated

g-C$_3$N$_4$. For the bulk g-C$_3$N$_4$ catalyst, a strong PL emission band is observed at 458 nm, resulting from the direct interband electron-hole recombination. A second weaker emission band observed at approximately 525 nm can be attributed to the radiative recombination of charge carriers captured by traps (structural defects) [15,33]. The spectrum of exfoliated g-C$_3$N$_4$ shows a blue shift, an increase in intensity, and a strong PL emission band at 436 nm. The blue shift of both bandgap and PL spectrum is most likely due to the quantum confinement effect, with the conduction and valence band shifting in opposite directions. Another possible reason is the reduction in conjugation length caused by the ultrathin structure of exfoliated g-C$_3$N$_4$ [34–36].

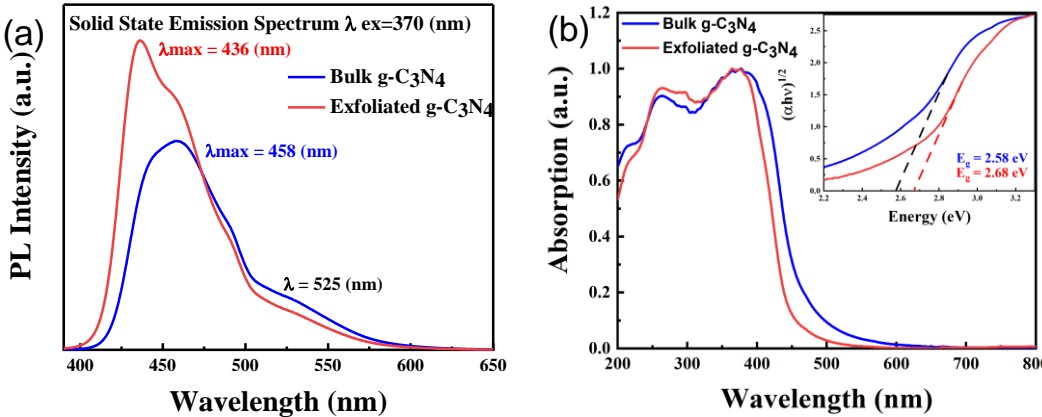

**Figure 3.** Photoluminescence spectra (**a**) Ultraviolet-visible absorption spectra (**b**) of bulk and exfoliated g-C$_3$N$_4$; the inset of (b) shows the Tauc plots.

Figure 3b shows the UV–vis absorption spectra and their respective Tauc plots (inset) of both materials. The absorption edges of exfoliated and bulk g-C$_3$N$_4$ were estimated at 440 and 460 nm, respectively, indicating visible light response and a blue shift in the absorption edge from bulk to exfoliated g-C$_3$N$_4$. The general tauc plot equation used for band gap calculation is as follows:

$$(\alpha h v)^n = A\left(h v - E_g\right) \tag{1}$$

Here, $\alpha$ is the absorption coefficient, $h v$ is the incident photon energy, $A$ is the constant, $E_g$ is the band and n is the nature of transition. A standard value of n equal to 1/2 is preferable for graphitic g-C$_3$N$_4$, which is ideal for an indirect bandgap semiconductor. In Tauc plot method, $h v$ is plotted on x-axis against $(\alpha h v)$ n on y-axis. Then, by extrapolating the linear part of the curve to a point where it intersects the x-axis is the optical band gap energy of the material. A slight increase in the bandgap of bulk g-C$_3$N$_4$ from 2.58 to 2.68 eV was observed after the exfoliation [31,37], which can be attributed to the quantum confinement effect [34,36].

The chemical state and composition of bulk and exfoliated g-C$_3$N$_4$ were analyzed by XPS, and the spectra corresponding to N1s and C1s are presented in Figure 4. The C1s spectra (Figure 4 a,b) of the samples show two strong peaks at 287.8 and 284.7 eV and two weak peaks at 286.2 and 293.5 eV. The weak peak at 293.5 eV corresponds to the π electron delocalization in C$_3$N$_4$ heterocycles, and that at 286.2 eV is attributable to amino functional groups (C–NH$_2$) bonded to the surface of g-C$_3$N$_4$. Meanwhile, the two strong peaks can be attributed to standard carbon bonds (C–C) and sp$^2$-hybridized carbon (N–C=N) in triazine rings [26,38,39]. As can be seen in Figure 4 (c,d), the N1s spectra of the samples show several peaks. The strong peak at 397.8 eV corresponds to sp$^2$ nitrogen bonded to carbon (C–N–C) of triazine rings. The peaks of medium intensity at 399.1 and 400.1 eV can be attributed to the presence of tertiary N in N–(C)$_3$ units and amino functional groups (NH$_2$ or NH). The weak peak observed at 403.5 eV can be assigned to positive charge localization in the cyano group and heterocycles and/or NO$_2$ terminal groups [15,40–42]. As shown

in Figure 4, the binding energy of C1s and N1s core electrons does not shift significantly, which suggests that the chemical states of both carbon and nitrogen in the exfoliated are the same as in the bulk g-$C_3N_4$ [43].

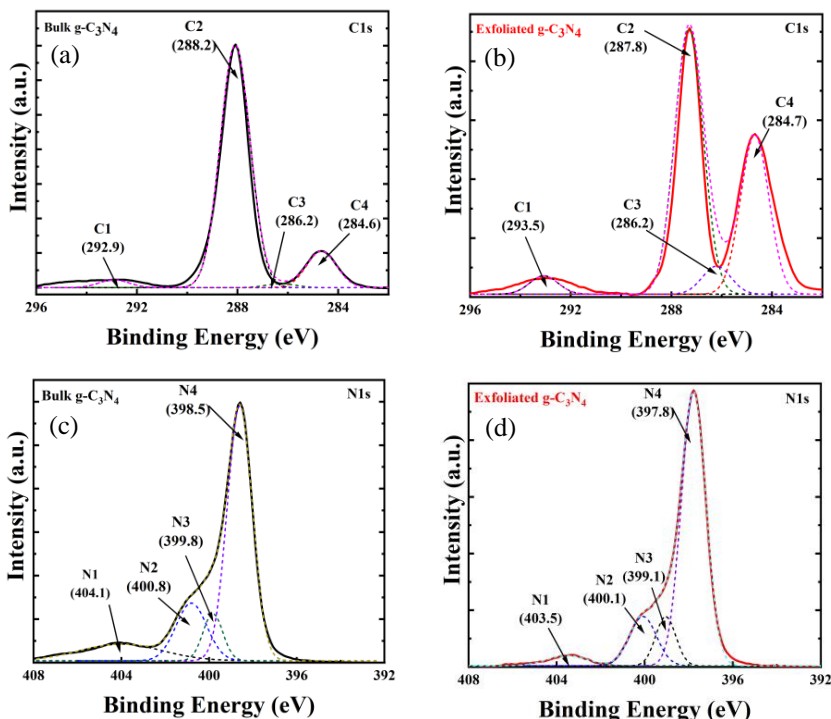

**Figure 4.** X-ray photoelectron spectra of bulk (**a**,**c**) and exfoliated (**b**,**d**) g-$C_3N_4$ (C1s and N1s).

The catalysts were then subjected to XRD analysis to examine the phase structures of g-$C_3N_4$ powders. As can be seen in Figure 5, the patterns of both bulk and exfoliated materials show a strong peak at 2θ = 27.2° and a weaker peak at 2θ = 13°. The strong peak can be indexed to the (002) plane, which is a characteristic interlayer stacking peak of aromatic g-$C_3N_4$ systems. Moreover, the weak peak, which can be assigned to the (100) crystal plane, is attributable to repeated units of tri-*s*-triazine [15,34]. The decrease in the intensity of the diffraction peaks for the treated materials indicates a modification of the interlayer structures [44,45].

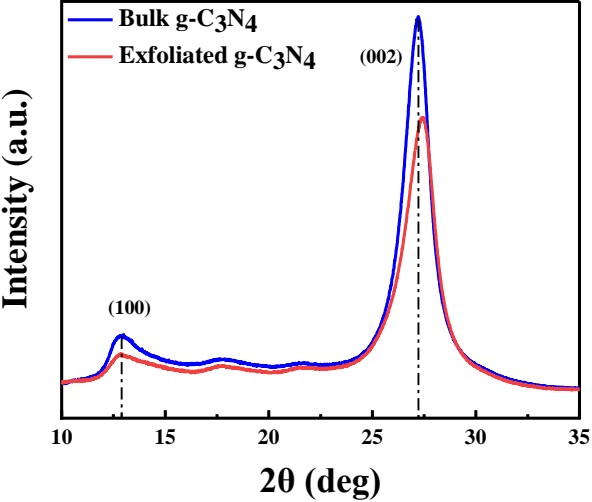

**Figure 5.** X-ray diffraction patterns of bulk and exfoliated g-$C_3N_4$.

The pH had a significant effect on the zeta potential of exfoliated g-$C_3N_4$. This effect was studied, and the results are shown in supplementary Figure S2. The possible ionization of primary, secondary, and tertiary amine groups on the surface of g-$C_3N_4$ can be described by the following equations [46]:

$$\equiv C - NH_2 + H^+ \rightarrow \equiv C - NH_3^+ \tag{2}$$

$$\equiv C - NH_2 + OH^- \rightarrow \equiv C - NH^- + H_2O \tag{3}$$

$$(\equiv C)_2 - NH + H^+ \rightarrow (\equiv C)_2 - NH_2^+ \tag{4}$$

$$(\equiv C)_2 - NH + OH^- \rightarrow (\equiv C)_2 - N^- + H_2O \tag{5}$$

$$(\equiv C)_3 - N + H^+ \rightarrow (\equiv C)_3 - NH^+ \tag{6}$$

Supplementary Figure S3 shows that the material surface is negatively charged in basic (pH = 10) and natural solutions (pH = 6.5) and positively charged at acidic pH (pH = 3), which is in accordance with Equations (2)–(6). The excess of protons $H^+$ in the system changes the overall charge on the surface to positive [46,47].

The bulk g-$C_3N_4$ synthesized has less surface area and aggregate like structure which is converted to nanosheets and there is an increase in the surface area as the result of exfoliation. The base characteristics of g-$C_3N_4$ remained the same, e.g., chemical structure and functional groups.

### 2.2. Photocatalytic Degradation of Phenol

The photocatalytic activity of the synthesized materials was studied using phenol as a model compound. The effect of the exfoliation process, light source, catalyst and pollutant concentration, pH, and dissolved oxygen on the photodegradation of phenol was studied in detail. The photolysis experiments were carried out without catalyst. All other experiments were carried out with water, phenol and catalyst reaction mixture. The homogeneous dispersion of the catalyst was ensured by sonicating the reaction mixture (water solution of phenol and catalyst), followed by stirring for 30 min in the dark. The photolysis experiments were conducted only under the influence of light without any catalyst. The reaction starts as soon as the light is turned on, before that there is only adsorption process.

### 2.2.1. Effect of Exfoliation

The photodegradation of phenol (initial concentration of 20 ppm) using 0.5 g/L bulk or exfoliated g-$C_3N_4$ was evaluated, and the results are displayed and compared in Figure 6a. The photolysis of phenol curve is presented, it shows that in presence of light and without catalyst there is almost no degradation. Bulk g-$C_3N_4$ showed approximately 40% phenol degradation after 180 min, whereas complete phenol degradation was achieved in 90 min of reaction time with exfoliated g-$C_3N_4$. As the bandgap of both materials is similar, the better photo efficiency of exfoliated g-$C_3N_4$ is most likely due to its higher surface area, which provides more available active sites compared with bulk g-$C_3N_4$. The obtained results are in accordance with the results reported by Babu, Bathula et al. and Ding, Wang et al. [48,49]. Due to its better photoefficiency, exfoliated g-$C_3N_4$ was used for further experiments.

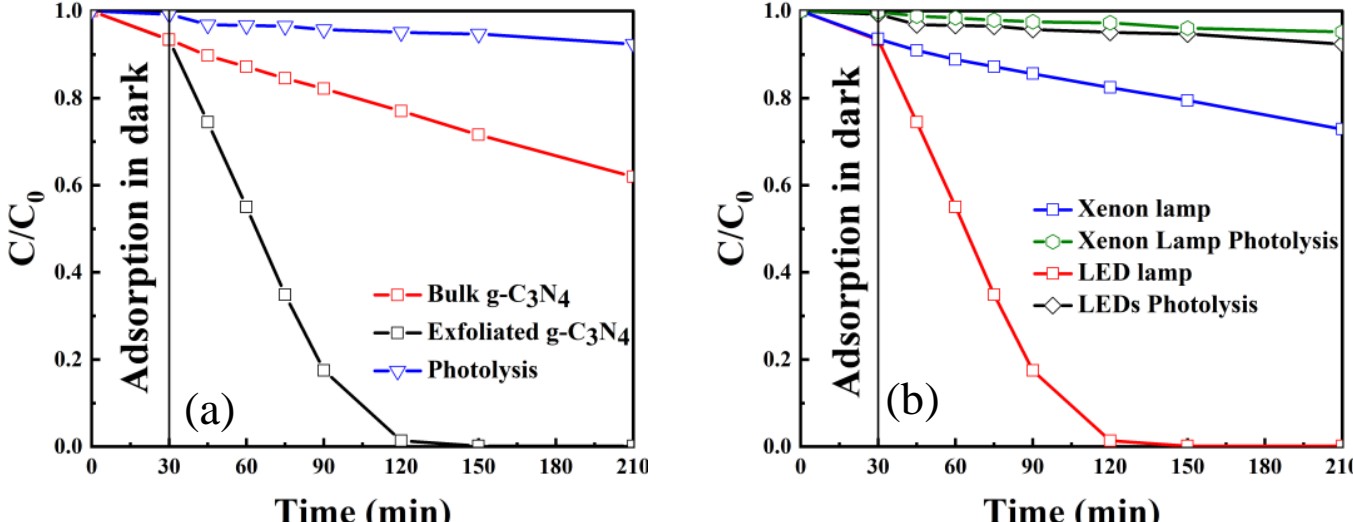

**Figure 6.** Phenol degradation with (**a**) bulk and exfoliated g-C$_3$N$_4$ (**b**) as a function of the light source; C$_0$ = 20 ppm, airflow = 50 mL/min, catalyst concentration = 0.5 g/L, pH = natural.

### 2.2.2. Effect of the Light Source

To study the effect of light on the photodegradation of phenol, two light sources were used, namely, a full-spectrum xenon lamp and a single-wavelength LED lamp. Figure 6b presents the performance of exfoliated g-C$_3$N$_4$ using the two different light sources. Experiments performed with both light source in the absence of a catalyst have shown no degradation of phenol. In this study, we used a custom-made immersion lamp with six LEDs of 430 nm because that was the maximum excitation wavelength of exfoliated g-C$_3$N$_4$ according to the PL spectra (Figure 3). A significant difference between the photodegradation using the two lamps was observed; namely, approx. 100% of phenol degradation was achieved within 90 min of reaction time using the LED lamp, whereas only 20% of degradation was achieved with the xenon lamp after 180 min. The faster reaction with the LED lamp is attributable to the higher availability of electrons at the maximum excitation wavelength. Therefore, exfoliated g-C$_3$N$_4$ and the LED lamp were selected for further experiments. The photocatalytic results of both lamps are compared to the photolysis as well to show that the degradation is due to the presence of a catalyst.

### 2.2.3. Effect of Catalyst Amount

The amount of catalyst was optimized to avoid using an excess of the catalyst and to enhance the efficiency of the degradation process. The amount of exfoliated g-C$_3$N$_4$ was varied from 0.1 to 1.5 g/L. Figure 7a shows the obtained phenol degradation curves. The degradation of phenol was negligible in the absence of the catalyst, and no significant improvement was achieved for catalyst concentrations below 0.5 g/L. By contrast, phenol was completely degraded in 90 min of reaction time for all the amounts above 0.5 g/L. Further increase in the catalyst amount resulted in a decrease in efficiency because of hindered light absorption and light scattering. Therefore, a catalyst concentration of 0.5 g/L was selected for further studies.

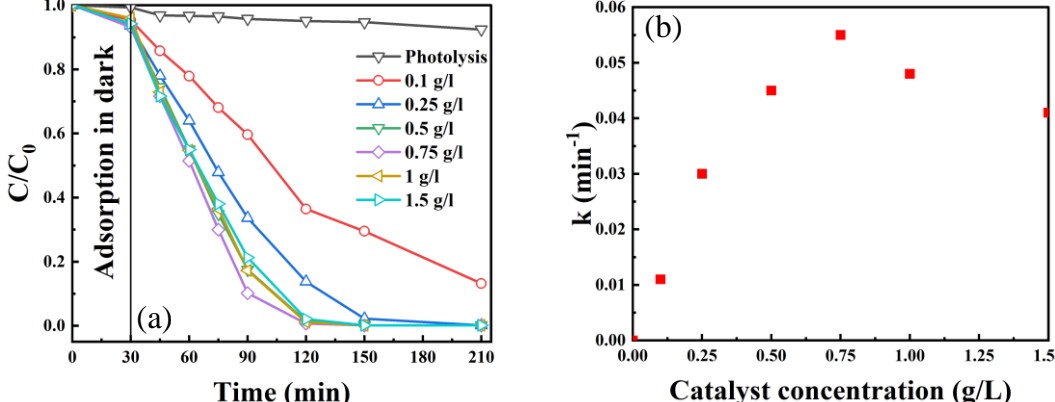

**Figure 7.** (**a**) Photodegradation curves of phenol concentration (**b**) reaction rate constant as a function of the catalyst concentration; $C_0$ = 20 ppm, airflow = 50 mL/min, pH = natural, light source = LEDs, Catalyst = exfoliated g-$C_3N_4$.

The photodegradation curves followed the exponential decay model and were fitted to obtain the reaction rate constant (k) [50,51] as follows:

$$\frac{C}{C_o} = e^{-kt} \tag{7}$$

The highest rate constant of 0.055 min$^{-1}$ was obtained for a concentration of 0.75 g/L (Figure 7b), whereas lower and similar values were found for 0.5 and 1 g/L, respectively. This decrease in the rate constant can be attributed to the increase in the catalyst amount hindering the absorption of light.

### 2.2.4. Effect of Pollutant Concentration

Figure 8a displays the phenol degradation curves obtained for different concentrations of phenol in an aqueous solution (20, 40, 60, 80, and 100 ppm) and the same amount of exfoliated g-$C_3N_4$. The overall efficiency of the process decreased as the pollutant concentration increased, which could be due to the excessive amount of pollutants covering the surface, with the concomitant decrease in photon absorption and the number of available active sites [52]. The dependence of the removal efficiency on initial phenol concentration was also studied and reported in terms of rate of reaction, expressed by the reaction rate constant. As shown in Figure 8b, and similar to data reported in the literature [53], the rate decreases with increasing initial phenol concentration.

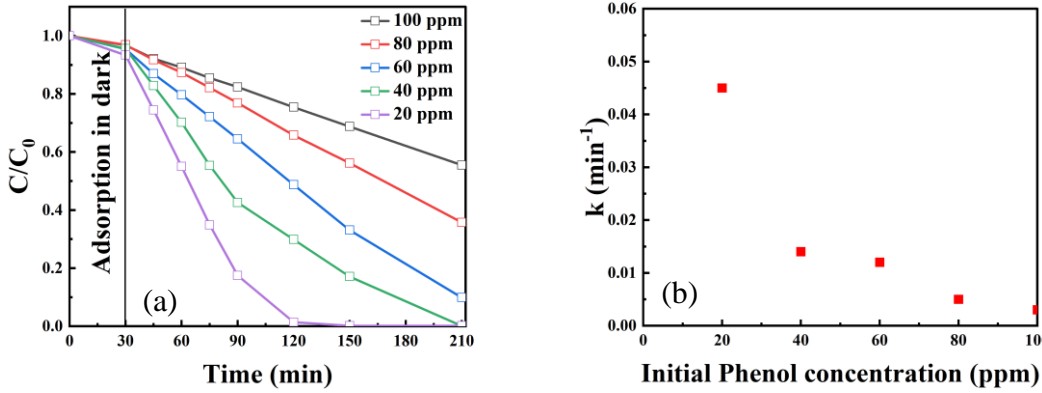

**Figure 8.** (**a**) Photodegradation curves of phenol as a function of the initial concentration; (**b**) reaction rate constant as a function of the initial phenol concentration airflow = 50 mL/min, catalyst = 0.5 g/L, pH = natural, light source = LEDs, Catalyst = exfoliated g-$C_3N_4$.

### 2.2.5. Effect of Initial pH

Figure 9 shows the influence of pH on phenol degradation. The photodegradation efficiency of phenol at acidic pH (2–5) was better than at basic pH (9–12). Almost 97% of phenol degradation was achieved in 60 min of reaction time at acidic pH compared with 100% for natural pH in 90 min and 80% at basic pH in 180 min, a similar pH dependency was shown in the literature [47,54]. The reason for this trend is that the catalyst surface is positively charged at acidic pH (see Figure S3), which enhances the adsorption of negatively charged molecules, e.g., $OH^-$, from the suspension compared with the negatively charged surface at basic pH. Hence, high pH is not favorable for the photodegradation of phenol [55,56].

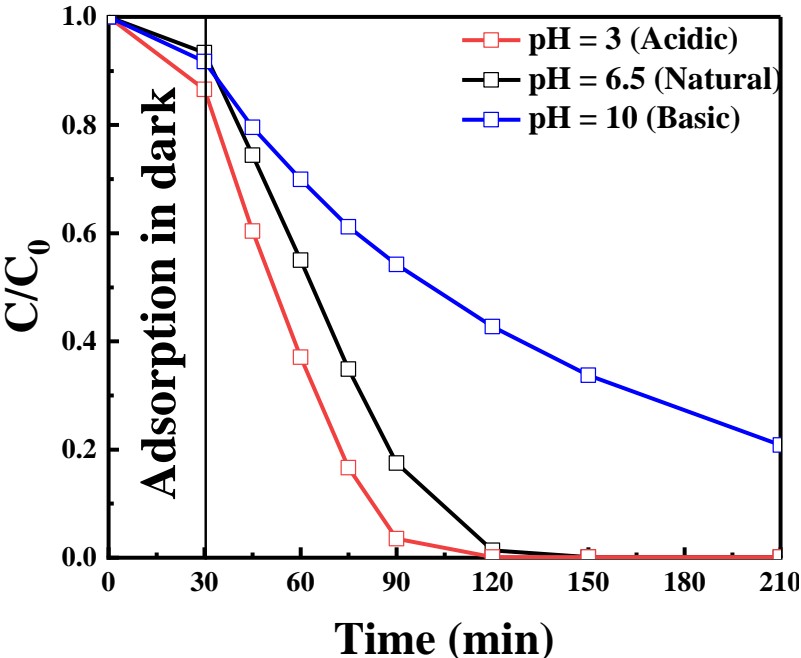

**Figure 9.** Photodegradation curves of phenol as a function of pH; C0 = 20 ppm, airflow = 50 mL/min, catalyst = 0.5 g/L, light source = LEDs, Catalyst = exfoliated g-$C_3N_4$.

### 2.2.6. Effect of Dissolved Oxygen

The influence of dissolved oxygen on the degradation of phenol was studied by conducting experiments with and without an external airflow. As can be seen in Figure 10, in both cases, the degradation curves show the same trend for the first 15 min. After that, in the absence of airflow, the phenol degradation slows down, probably because of the consumption of all dissolved oxygen in the aqueous solution, and no change in the degradation is observed after the first 30 min. During the reaction in the absence of air, the system is closed and there is much less overhead space above the reaction medium in the reactor for enough air to carry out the reaction. By contrast, complete degradation is observed under an airflow within 90 min of reaction time because of the presence of extra oxygen to perform the reaction [54]. Oxygen plays an important role in trapping the photoinduced electron produced by the photoactivation of catalysts, consequently reducing the possibility of electron/hole ($e-/h+$) recombination as well. [57,58]. The mechanisms proposed in Equations (8–12) are for activation of catalyst by light and formation of $H_2O_2$. In several studies, the authors suggested that the catalyst is activated with light (*hv*) which originates the electron/hole ($e-/h+$) pairs [10,57,58]. $H_2O_2$ is formed during the process by the utilization of the electrons produced and oxygen present in the system. Then, $H_2O_2$ and $O_2$ are reduced to $OH^-$ and superoxide radical $O_2^-$ which are used in the degradation of organic pollutants.

$$photocatalyst + h\nu \; \rightarrow \; e^- + h^+ \tag{8}$$

$$O_2 + e^- \; \rightarrow \; O_2^- \tag{9}$$

$$O_2 + 2e^- + 2H^+ \; \rightarrow \; H_2O_2 \tag{10}$$

$$H_2O_2 + e^- \; \rightarrow OH^- + OH^* \tag{11}$$

$$OH^- + organic\ pollutant \; \rightarrow intermediates + CO_2 + H_2O \tag{12}$$

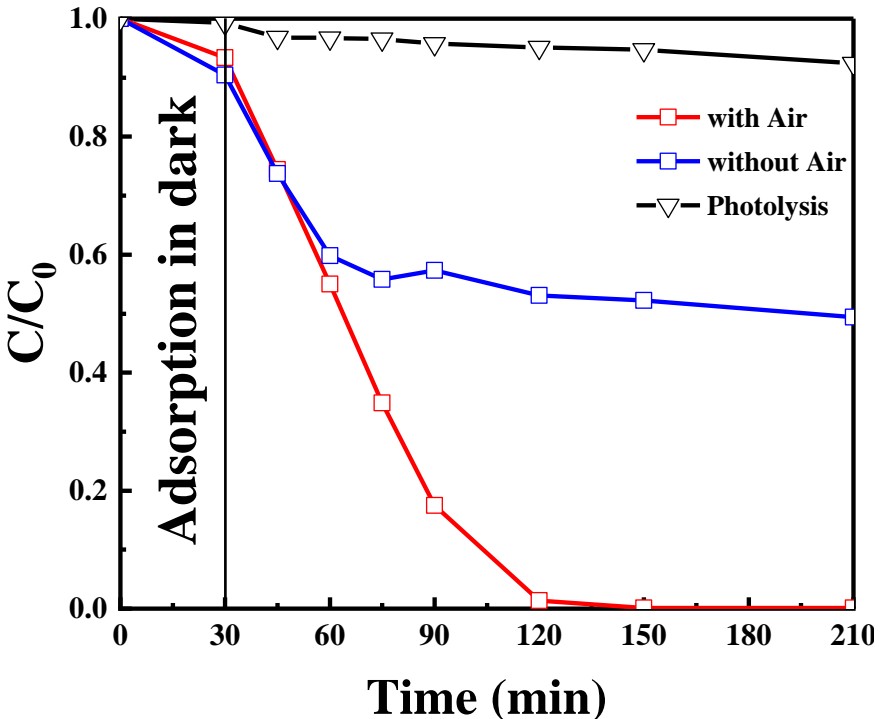

**Figure 10.** Photodegradation curves of phenol in the absence or presence of an airflow; $C_0$ = 20 ppm, airflow = 50 mL/min, catalyst = 0.5 g/L, pH = natural, light source = LEDs, Catalyst = exfoliated g-$C_3N_4$.

## 2.2.7. Effect of $TiO_2$ and its Composite

Titanium oxide ($TiO_2$) is one of the most prominent and widely used semiconductor photocatalysts. The performance of g-$C_3N_4$ was compared with that of $TiO_2$ and a composite of both materials, i.e., exfoliated g-$C_3N_4$:$TiO_2$ (1:1). Figure 11 shows the phenol degradation curves obtained under the same operating conditions. Exfoliated g-$C_3N_4$ afforded complete phenol degradation in 90 min of reaction time, $TiO_2$ showed 50% degradation in 180 min, and complete degradation was also obtained with the composite in 150 min. The better performance of exfoliated g-$C_3N_4$ can be attributed to the same excitation wavelength of exfoliated g-$C_3N_4$ and the LED lamp (430 nm), whereas the maximum absorption of $TiO_2$ lies in the range of 190–400 nm [59]. Consequently, the addition of exfoliated g-$C_3N_4$ to $TiO_2$ enhanced its performance under visible light irradiation.

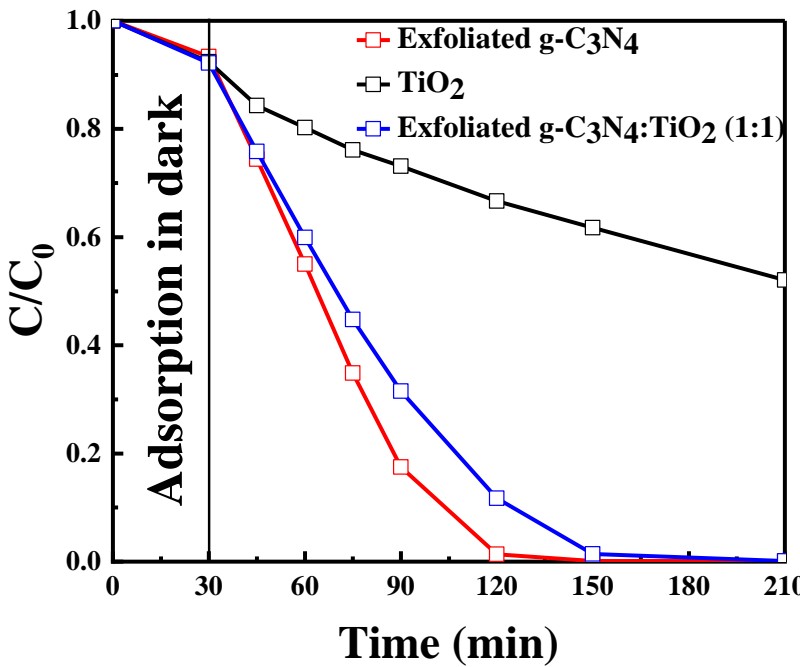

**Figure 11.** Phenol degradation with exfoliated *g*-C$_3$N$_4$, TiO$_2$, and a *g*-C$_3$N$_4$:TiO$_2$ composite; C$_0$ = 20 ppm, airflow = 50 mL/min, catalyst = 0.5 g/L, pH = natural, light source = LEDs, Catalyst = exfoliated g-C$_3$N$_4$.

### 2.2.8. Catalyst Reutilization Tests

The reusability of exfoliated g-C$_3$N$_4$ was demonstrated by performing three cycles of phenol degradation after subjecting the catalyst to thorough washing and drying at 120 °C for 12 h. The degradation efficiency of the catalyst remained unchanged for at least three cycles (supplementary Figure S3), and the complete degradation of phenol was achieved within 3 h. It can be concluded that the material remained stable for at least three cycles while maintaining its photocatalytic activity.

### 2.2.9. Photocatalytic Efficiency for the Degradation of a Mixture of Organic Pollutants

Three additional phenolic compounds commonly found in oil refinery wastewater [60], namely, *m*-cresol, catechol, and xylenol, (chemical structures in supplementary Table S1) were chosen to further study the photocatalytic degradation performance of exfoliated g-C$_3$N$_4$. Figure 12 shows the photodegradation curves for each organic pollutant in single-compound aqueous solutions. Xylenol was completely degraded in approximately 60 min, phenol and *m*-cresol in 120 min, and catechol in 150 min of reaction time. In the photocatalytic reactions of aromatic compounds, the degradation is considered to be initiated by an electrophilic attack of the "holes" produced by exfoliated g-C$_3$N$_4$ to the benzene ring. Hydroxyl (–OH) and alkyl (e.g., –CH$_3$) groups are generally known to increase the electron density of the aromatic ring, making it more susceptible to undergo an electrophilic attack at its *ortho* and *para* positions. Compared with phenol, catechol was more difficult to degrade. Even though catechol has two–OH groups, their *ortho* orientation reduces the number of distinct carbon atoms that are prone to undergo electrophilic substitution [10,61]. The *meta* orientation of the –CH$_3$ and–OH groups in *m*-cresol is favorable to a facile attack on the aromatic ring in three distinct positions, providing *m*-cresol with a higher reactivity compared with phenol. Having the highest number of activating substituents (–OH and 2 × –CH$_3$), the aromatic ring of xylenol degraded the fastest among the four studied phenolic compounds.

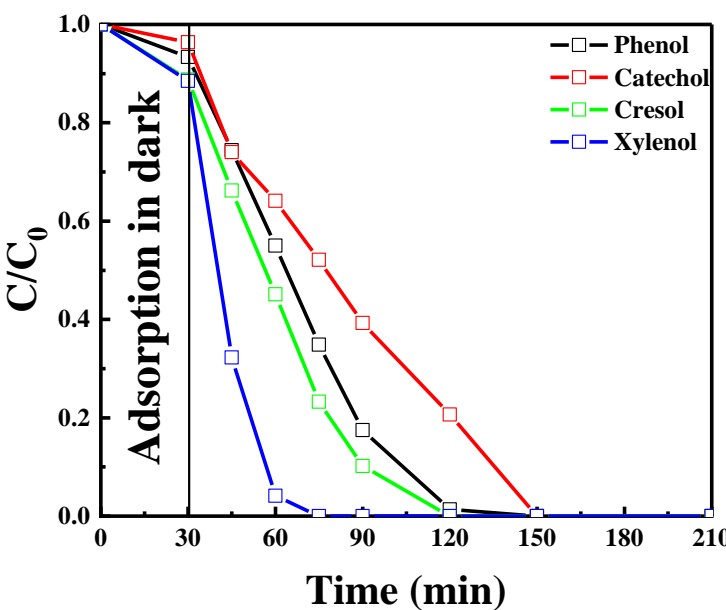

**Figure 12.** Degradation profile of (**a**) individual pollutants (20 ppm) and (**b**) a mixture of 20 ppm of each pollutant; $C_0$ = 20 ppm, airflow = 50 mL/min, catalyst = 0.5 g/L, pH = natural, light source = LEDs, Catalyst = exfoliated g-$C_3N_4$.

## 3. Materials and Methods

### 3.1. Chemicals and Materials

Melamine ($C_3H_6N_6$, 99%), *m*-cresol ($CH_3C_6H_4(OH)$, 99%), and catechol ($C_6H_4(OH)_2$, 99%) (Alfa Aesar); phenol ($C_6H_5OH$, 99%) (Merck); xylenol (2,6-dimethylphenol) (($CH_3)_2C_6H_3OH$, 99%) and titanium oxide ($TiO_2$) (Acros); hydrochloric acid (HCl), sodium hydroxide (NaOH) (VWR); acetonitrile ($C_2H_3N$, 99.99%) and ultrapure water for high-performance liquid chromatography (HPLC) (Sigma Aldrich) were used as received without any pretreatment.

### 3.2. Synthesis and Characterization of the Catalysts

Bulk g-$C_3N_4$ was prepared by thermal decomposition of melamine. The precursor (melamine) 5g was placed in a closed crucible in a muffle furnace (Carbolite, GPC 1200) under a static air atmosphere; 2.6 g g-$C_3N_4$ were produced which corresponds to a yield of 52%. A heating ramp of 2 °C min$^{-1}$ was programmed up to 450 °C, and this temperature was maintained for 2 h. Next, the temperature was increased up to 550 °C using a heating ramp of 2 °C min$^{-1}$ and then maintained for 4 h. After cooling down naturally, the material was crushed to powder using a mortar and pestle, rinsed with ultrapure water, filtered, and dried overnight at 80 °C [15]. The exfoliation of the prepared catalyst was performed by placing the catalyst in an open crucible inside a muffle furnace for 2 h at 500 °C using a heating ramp of 2 °C min$^{-1}$. The g-$C_3N_4$ material obtained after exfoliation is referred to hereafter as exfoliated g-$C_3N_4$. Two catalysts, namely, bulk g-$C_3N_4$ and exfoliated g-$C_3N_4$, were characterized and tested in photodegradation experiments. The molecular structure of the g-$C_3N_4$ is presented in supplementary materials (Figure S4). Additionally, the performances of commercial $TiO_2$ and its composite with g-$C_3N_4$ ($TiO_2$:g-$C_3N_4$), which were prepared in this study, were examined and compared with that of exfoliated g-$C_3N_4$. Briefly, an equal amount of exfoliated g-$C_3N_4$ and commercial $TiO_2$ were mixed separately in methanol and sonicated, then the two mixtures were mixed and again sonicated for 6 h. Later the mixture was centrifuged and the solid was dried overnight at 80 °C.

Brunauer–Emmett–Teller (BET) surface area, pore-volume, and pore size distribution were determined using an autosorb ®iQ-MP/XR (Nova 4200e), Quantachrome instrument (Germany). The samples were degassed at 120 °C for 3 h before the analysis. The surface area of the catalyst was calculated from $N_2$ adsorption-desorption isotherms at −196 °C.

The Barrett–Joyner–Halenda (BJH) method was used to determine the micropore volume. UV–vis absorption spectra were recorded using a UV-3600 Plus (Shimadzu, Kyoto, Japan) spectrometer with medium scan speed using a 20 slit width with three external detectors in the range of 200–800 nm. Solid-state photoluminescence (PL) spectra were obtained using an FP-8300 (Jasco, Hachioji-shi, Japan) spectrofluorometer with a 150 W xenon lamp as a light source. The measurements were performed using both excitation and emission bandwidth fixed at 2.5 and 5 nm, respectively, with low sensitivity. The excitation wavelength was set at 370 nm, and the emission was measured in the range of 380–700 nm. A ZEN5600 Zetasizer Nano (Malvern, UK) instrument was used to measure the zeta potential of the synthesized materials, which was determined using phenol and catalyst aqueous suspensions at acidic, basic, and natural pH. Hydrochloric acid and sodium hydroxide were used to adjust the pH of the suspensions. To investigate the crystalline phases of the synthesized catalysts, X-ray diffraction (XRD) analysis was conducted with a Mini Flex 600C (Rigaku, Tokyo, Japan) diffractometer using CuK$\alpha$ radiation at a voltage of 40 kV, a current of 15 mA, and a spin speed of 80 rpm in the range between 3° and 60° with a step size of 0.0075°. Transmission electron microscopy (TEM) images were obtained using a JEM 1400 plus (JEOL, Akishima, Japan) microscope (LaB$_6$ filament) using Quantifoil S7/2 (2 nm carbon film) grids and a CCD camera (Ruby) at 120 kV. X-ray photoelectron spectroscopy (XPS) was conducted using a Thermo Scientific (Waltham, MA, USA) K Alpha and X-ray Photoelectron Spectrometer. All samples were analyzed using a micro-focused, monochromated Al K $\alpha$ X-ray source (1486.68 eV; 400 μm spot size). The analyzer had pass energy of 200 eV (survey), and 50 eV (high-resolution spectra), respectively. To prevent any localized charge buildup during analysis the K-alpha +ve charge compensation system was employed at all measurements. The samples were mounted on conductive carbon tape the resulting spectra analyzed using the Avantage software from Thermo Scientific.

### 3.3. Photocatalytic Experimentation

The photocatalytic experiments were conducted in a jacketed glass reactor (Peschl Ultraviolet GmbH) with a maximum working capacity of 225 mL, under a custom-made LED immersion lamp comprising 6 LEDs (each of 10 W) with maximum emission at 430 nm and UV-visible light irradiation using a 150 W full-spectrum (200 nm to 800nm) xenon lamp. The reactor was placed in a safety cabinet (Supplementary Figure S1).

The homogeneous dispersion of the catalyst was ensured by sonicating the reaction mixture (water solution of phenol and catalyst), followed by stirring for 30 min in the dark under a continuous airflow to maintain the adsorption-desorption equilibrium. The photolysis experiments were conducted only under the influence of light without any catalyst. In a typical experiment, the initial concentration of phenol was kept at 20 ppm, and the catalyst concentration was varied from 0.1 to 1.5 g/L in a 225 mL phenol solution. Samples (1 mL) were collected periodically from the reaction mixture for the determination of the concentration of phenol. After centrifugation and filtration, the samples were analyzed by HPLC. Experiments were also conducted to study the effect of process parameters, including pH, catalyst concentration, pollutant concentration, air, and exfoliation, on the performance of the catalysts. Experiments with other phenolic compounds, such as *m*-cresol, catechol, and xylenol, were conducted using 0.5 g/L catalysts and 20 ppm pollutant concentration at natural pH. The catalysts exhibited reusability properties, which were established by performing three cycles of reutilization test. The catalysts were tested for reutilization after thorough washing with water and drying at 120 °C for 12 h. The percentage of degradation efficiency was determined using the following Equation (13):

$$Degradation\ Efficiency = \frac{C_o - C}{C_o} \times 100 \qquad (13)$$

where, $C_o$ (mg/L) and $C$ (mg/L) are the initial phenol concentration and the residual phenol concentration in the solution at an irradiation time t, respectively. The rate of the reaction of the experiments were calculated using Equation (7), the rate is determined

for the time of the reaction in which either the lights are on and reaction is taking place or there is change in the concentration of the pollutant. $C/C_o$ was plotted against time and the exponential model (Equation (7)) was fitted to the data, and the slope of the curve is the rate constant "k" of the single reaction. Different reaction rate constants were plotted in Figures 7b and 8b. The concentration of phenol was analyzed using a Prominence HPLC system (Shimadzu, Japan) comprising a binary pump (LC-20AB), a SIL-20A autosampler, a DGU-20A3 degasser, and an SPD-M20A diode-array detector. The analysis was conducted with a Phenomenex (C18, 150 mm × 4.6 mm, 3 μm) column at a fixed flow rate of 0.8 mL/min, using water (A) and acetonitrile (B) as a mobile phase with the following gradient: 15% B followed by 60% B in 7 min and back to 15% B in 8 min (injection 5 μL; UV 254 nm). Phenol and xylenol were analyzed at a maximum absorption wavelength ($\lambda_{max}$) of 270 nm, catechol at 277 nm, and *m*-cresol at 273 nm.

## 4. Conclusions

In this study, g-$C_3N_4$ was synthesized using melamine as a precursor and exfoliated. The morphology, structure, and optical and surface properties of the material were characterized using TEM, XRD, XPS, PL, and BET. The chemical structure of g-$C_3N_4$ was confirmed by its characteristic peaks in the FTIR and XPS spectra, and its phases were identified by XRD. The zeta potential was measured at different pH values, which confirmed that surface charges of g-$C_3N_4$ particles are caused by different surface groups. Phenol was used as a model pollutant to study the influence of process parameters on photocatalytic degradation. It was found that the exfoliation significantly improves the optical (absorption) and surface properties (surface area, porosity) of bulk g-$C_3N_4$, resulting in enhanced photocatalytic performance in comparison with that of non-exfoliated g-$C_3N_4$. The use of a visible light single-wavelength LED lamp as an irradiation source improves the photocatalytic degradation of phenol compared with a full-spectrum xenon lamp. The degradation rate increases with the catalyst concentration, whereas it decreases in the presence of the excess amount of pollutants (i.e., phenol). Moreover, it was shown that an acidic environment is favorable for the photocatalytic degradation of phenol. Dissolved oxygen proved to be an important factor in the degradation of phenol since only 50% of the degradation was achieved in the absence of oxygen. A comparison between g-$C_3N_4$ and $TiO_2$ showed that the performance of the former in the degradation of phenol under visible light irradiation is better. Furthermore, $TiO_2$ performance improves with the addition of g-$C_3N_4$. The versatility of exfoliated g-$C_3N_4$ was demonstrated in the degradation of other phenolic pollutants individually and in a mixture. Furthermore, exfoliated g-$C_3N_4$ proved to be a stable catalyst that can be recycled and reused without any loss of activity.

**Supplementary Materials:** The following are available online at https://www.mdpi.com/article/10.3390/catal11060662/s1, Figure S1 Photocatalytic reactor setup, Figure S2 Zeta potential of exfoliated g-C3N4 as a function of the pH value of the suspension, Figure S3 Phenol degradation conversion for the reutilized catalyst; $C_0$ = 20 ppm, airflow = 50 mL/min, catalyst = 0.5 g/L, pH = natural, Table S1 Chemical structures of the pollutants used.

**Author Contributions:** Conceptualization, A.G.R.; Formal analysis, A.G.R. and M.T.; Investigation, A.G.R. and M.T.; Methodology, A.G.R.; Resources, M.M.; Supervision, M.M.; Validation, A.G.R.; Visualization, A.G.R.; Writing original draft, A.G.R.; Writing review and editing, M.S. and M.M. All authors have read and agreed to the published version of the manuscript.

**Funding:** This research received no external funding.

**Acknowledgments:** Adeem Ghaffar Rana acknowledges the financial support from the Higher Education Commission, Pakistan, and Deutscher Akademischer Austauschdienst (DAAD), Germany.

**Conflicts of Interest:** The authors declare no conflict of interest.

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
