# Peer review of "Efficient Advanced Oxidation Process (AOP) for Photocatalytic Contaminant Degradation Using Exfoliated Metal-Free Graphitic Carbon Nitride and Visible Light-Emitting Diodes"

_catalysts, doi:10.3390/catal11060662_

Round 1

Reviewer 1 Report

The authors’ comparative photocatalytic studies on bulk graphitic carbon nitride and exfoliated g-C3N4 are very interesting and definitely publishable. However, authors must pay attention to details while describing the results. For example, figure 6, 7 and 10 contain a graph designated as photolysis and no explanation was given. Is it a blank photolysis or the photolysis of phenol without a catalyst? This needs to be explained both in the method and the figures. For a photocatalytic experiment, it is not clear what was the solvent system used to run each experiment. It will be easier readers to follow for if the authors describe the methods related to each figure in separate paragraphs.

Additionally, the authors must address the following: -

  • Provide a crystal pattern or molecular structure of the graphitic carbon nitride. It will help the reader to see the repeated units of tri-s-triazine.
  • There are major discrepancies in figures 1, 2, 3 and 4 and corresponding descriptions in the ‘Results’. Fix it.
  • Line 140-144: It is not clear how the band gap was calculated. A mathematical formula and corresponding description will be useful.
  • Figure 6, figure 9: The authors claimed that the reactions were completed in 90 min while C/C0 were ~ 0.15 i.e., 15% incomplete. Rather, the reaction got completed in 120 mins. Similarly, in figure 9, it was described that 97% degradation was achieved in 60 minutes while C/C0 was ~ 0.35. Provide an appropriate explanation to describe these data.
  • It will be useful to include the UV-Vis traces of photocatalytic degradation of phenol. This will help understand if there is any secondary product formation. These side products must not be any pollutants, otherwise the purpose of this photocatalysis study won’t be met.
  • Line 235: Describe how the rate constant was calculated in ‘Method’. What is the meaning of highest rate constant? The rate determination method is not continuous here since there is periodic time lag to remove 1 mL sample and filter followed by HPLC analysis. It is important to be thorough while describing the rate constant and figure 7b.
  • The proposed mechanism is not accurate. What is the source of CO2? It should not be produced from phenol since the benzene ring cannot be cleaved under such conditions. The proposed mechanism must be revised from literature studies.
  • Line 57-59: provide references.
  • Line 166-171: provide appropriate references to support your claim.

Reviewer 2 Report

This paper deals with PC /photocatlytical/ activity of g-C3N4, together with a  characterization and preparation description. The experimental data are well supported by physical methods: PL, UV/Vis, TEM, N2-BET, XRD + PC activity experiments. The papers sounds well, and can be published AFTER MINOR REVISION. Some questions (Q) and remarks (R) are listed bellow, they are important for paper quality.

R1 the abstract must be shortened significantly

Q1 please, explain better in section Materials and Methods what  happens with the g-C3N4 during exfoliataion concerning structure, chemistry, sizes, texture properties, for  a better understanding of reader dealing with ceramic materials, deffect chemistry etc. 

R2 the objective of the study need a reformulation, especially the lamp description here can be moved to the Materials and Methods section!

R3 Fig. 2 a, the Y-Title is wrong, it must be R%, or F(R)

R4 Fig. 2 b, the X - title is wrong
